# Diversity of Termite Breeding Systems

**DOI:** 10.3390/insects10020052

**Published:** 2019-02-12

**Authors:** Edward L. Vargo

**Affiliations:** Department of Entomology, Texas A&M University, College Station, TX 77843, USA; ed.vargo@tamu.edu; Tel.: +1-979-845-5855

**Keywords:** reproductives, king, queen, neotenics, parthenogenesis

## Abstract

Termites are social insects that live in colonies headed by reproductive castes. The breeding system is defined by the number of reproductive individuals in a colony and the castes to which they belong. There is tremendous variation in the breeding system of termites both within and among species. The current state of our understanding of termite breeding systems is reviewed. Most termite colonies are founded by a primary (alate-derived) king and queen who mate and produce the other colony members. In some species, colonies continue throughout their life span as simple families headed by the original king and queen. In others, the primary king and queen are replaced by numerous neotenic (nymph- or worker-derived) reproductives, or less commonly primary reproductives, that are descendants of the original founding pair leading to inbreeding in the colony. In still others, colonies can have multiple unrelated reproductives due to either founding the colonies as groups or through colony fusion. More recently, parthenogenetic reproduction has shown to be important in some termite species and may be widespread. A major challenge in termite biology is to understand the ecological and evolutionary factors driving the variation in termite breeding systems.

## 1. Introduction

Like all eusocial insects, termites have a reproductive division of labor in which specialized castes reproduce and other members of the colony work on their behalf. For the purposes of this review, the breeding system is defined as the number and caste members engaged in reproduction within termite colonies. Here, breeding system and breeding structure are used interchangeably. The breeding system determines the degree of genetic relatedness among colony members, as well as the level of inbreeding of individuals within the colony. Therefore, understanding the breeding system of a species is critical for knowing both the direct and indirect fitness benefits that colony members experience.

The colonies of most termites are started by a pair of primary (alate-derived) reproductives following a mating flight. The primary reproductives, the king and queen, form a monogamous pair that produces the other colony members. In many termites, especially in the so-called lower termites, but also in some higher termites, neotenic individuals (derived from nymphs or workers), assume the reproductive role, usually after the death of the king and/or queen. Of course, there are many exceptions to this general rule, and recent studies combining field censusing and genetic analyses continue to add new twists to the breeding system of termite colonies. The breeding system of termites can be categorized as simple families, extended families, mixed families, or involving parthenogenetic reproduction. 

Termite breeding systems have been the subject of a couple of reviews. One by Vargo and Husseneder [1] focused on the subterranean genera, *Coptotermes* and *Reticulitermes*, and another focused on the colony and population genetics of termites [2]. Since these reviews, there have been some important developments in our understanding of termite breeding systems. In this review, I will present the current state of knowledge in the field. First, I review the various reproductive castes found in termites.

## 2. Reproductive Castes

There are several different castes in termites capable of reproduction. Not all castes are present in all species.

### 2.1. Primary Reproductive

The main reproductive castes are the primary male and female reproductives. These are the only true adults in the colony, with fully developed wings, compound eyes, and body pigmentation. Primary reproductives fly from the colony and then land at varying distance from the natal nest. Males (kings) and females (queens) pair up to form what is known as tandem pairs, then locate a nest site, mate and produce offspring. Primary reproductives are the main way that new colonies are formed.

### 2.2. Neotenic Reproductive

Neotenic reproductives develop from either nymphs or workers. Nymph-derived neotenics, known as brachypterous neotenics, have wing pads and often are slightly melanized. Worker-derived neotenics, called apterous neotenics, lack wing pads and show little if any melanization. The occurrence of neotenics throughout the termites is reviewed by Myles [3]. Neotenics are widespread among the lower termites where they occur in over 60% of the genera, whereas they occur in only about 13% of the higher termite (Termitidae) genera. Since neotenics are unable to fly, they stay in the nest where they inherit existing resources, including the colony worker force. They allow the colony to live beyond the lifespan of the primary reproductives but they inbreed within the nest. Neotenics tend to be less fecund than primary reproductives, but they often occur in large numbers and the combined reproductive output of the group can far exceed that of the primary reproductives. Neotenics are common in wood feeding termites but are rare in mound building and humivorous species.

### 2.3. Soldier Reproductive

In some lower termites, individuals of both sexes with soldier-like morphology have functional gonads and reproduce [4]. This occurs in four species of Archotermopsidae and two species of Stolotermitidae, but are especially common in *Archotermopsis wroughtoni* and *Zootermopsis nevadensis* [5,6].

## 3. Types of Breeding Systems

### 3.1. Simple Families

The colonies of most termites, at least during the founding stage and on through the early colony growth phase, are headed by a single pair of monogamous primary reproductives. Such colonies, in which colony members are the progeny of one king and one queen, are simple families. In some higher termites, colonies remain simple families throughout their life cycle. However, in many termites, including some higher termites and most lower termites, the breeding structure can change from a simple family through the addition of new reproductives, usually neotenics, or through colony fusion. Whether colonies change from simple families to some other breeding system and the point in the lifecycle when the change occurs differs tremendously among species. This variation within and among species underlies much of the diversity of breeding systems in termites. The selective forces favoring one breeding system over another are not well understood.

While most simple families are headed by primary reproductives, this may not always be the case. It is possible for a colony to be a simple family headed by two neotenic reproductives or one neotenic and one primary reproductive, as long as all the individuals in the colony are the offspring of the reproductive pair. Since neotenics cannot found colonies independently, for a simple family to be headed by one or two neotenics, the neotenics would need to reproduce for a long enough period that their offspring replace any offspring of the primary reproductive(s) they supersede. Given the relatively long life span of many termite workers and soldiers, such situations are most likely rare.

The proportion of colonies forming simple families within a population is highly variable both within and among species. For example, only simple family colonies were found in 17 colonies studied genetically in the Neotropical termitid *Labiotermes labralis* [7] and 38 colonies of the Australian subterranean termite *Coptotermes lacteus* [8], whereas about half of 13 colonies of the subterranean termite *R. malletei* were simple families and half were extended families [9].

In the case that a simple family is headed by a king and queen who are unrelated to each other, the colony will be outbred. In contrast, colonies headed by siblings will be inbred. The only way to tell the degree of relatedness of the reproductives is by genotyping them directly, or more commonly, inferring their genotypes from the genotypes of workers. Vargo and Husseneder [2] reviewed the studies that have inferred the degree of relatedness among the reproductives heading simple family colonies. There is considerable variation within and among species in the degree of relatedness between kings and queens. The reasons for this variation are unknown but are presumably related to the distance dispersed by alates from the natal colony. Where alates fly long distance, the chance that siblings pair up should be low, whereas short-range dispersers are more likely to pair with siblings.

The use of genetic markers can uncover breeding systems that may be difficult to obtain through field censuses. For example, in the most basal termite, *Mastotermes darwiniensis* (Mastotermitidae), the colonies are founded by primary reproductives, but only numerous neotenics have been found in excavations of established field colonies [10]. Goodisman and Crozier [11] conducted genetic analysis of 18 colonies of *M. darwiniesis* in northern Australia using microsatellite markers. They found that 26% of the colonies had genotypes consistent with a single monogamous pair of reproductives, while 47% had genotypes characteristic of multiple inbreeding neotenics derived from a monogamous pair of reproductives, and 27% had evidence of complex genotypes indicating the presence of offspring descended from multiple primary pairs, either from colonies founded by multiple pairs of primary reproductives or the fusion of colonies after establishment. Thus, the breeding system of *M. darwiniensis* is much more complicated when viewed through the lens of genetic data than through that of field collections.

### 3.2. Extended Families

Colonies headed by multiple secondary reproductives descended from a single monogamous pair are called extended families following the terminology of Vargo [12]. Most often, such colonies are headed by multiple neotenics (either worker or nymph-derived reproductives) that are descended from the primary king and queen. Neotenics are more common in the lower termites [3], so extended family colonies are expected to be more common in these taxa than in the Termitidae. However, in some termitids, secondary reproductives can be alate-derived primary reproductives recruited from within the colony [13], so extended family colonies are not restricted to the presence of neotenics.

Extended family colonies can be identified genetically using biparentally inherited markers such as microsatellites or allozymes. The genotypes of a pool of workers in extended family colonies have no more than four alleles at a locus, the maximum number possible from a mating of two individuals but have the presence of more genotypes than possible from a mating between a monogamous pair (e.g., an allele paired with itself and two other alleles). Examination of a mitochondrial DNA marker would show that all workers were identical since they all descended from the same original foundress queen.

By definition, extended family colonies are headed by interbreeding groups of closely related individuals leading to inbred colonies. The degree of inbreeding within a colony will depend on the number of breeders and the number of generations of inbreeding. In general, the fewer the numbers of breeders and the more generations they inbreed, the more inbred the colony will be. If there are many extended family colonies in a population, this can have a strong effect on the overall level of inbreeding in the population. In the genus *Reticulitermes*, the proportion of extended family colonies can vary from 5% in a population of *R. hageni* [14] to 100% in invasive populations of *R. flavipes* [15]. Within a species, there can be considerable variation in breeding system types. Figure 1 shows variation in the breeding system among populations of *R. flavipes* in the U.S.

Neotenics are produced in colonies in response to orphaning, i.e., the death of the primary king or queen. Recent work by Sun et al. [16] shows that there is both sex-specific inhibition and stimulation of neotenic development in *R. flavipes*. These authors show that the presence of neotenics of one sex inhibits workers of the same sex from developing into neotenics but also stimulates workers of the opposite sex to differentiate into neotenics. The production of neotenics has long been believed to be under the influence of pheromones [17], but it was not until recently that biologically active compounds were identified. Matsuura et al. [18] identified two volatile compounds, n-butyl-n-butyrate and 2-methyl-1-butanol, that are the active components in the queen pheromone that inhibits the development of female neotenics in *R. speratus*. These compounds are also produced by eggs [18] and inhibit egg laying among female neotenics [19]. So far, this is the only pheromone identified that inhibits the development of female neotenics and no pheromones have been identified that inhibit the development of male neotenics. Hayashi et al. [20] proposed a sex-linked genetic mechanism underlying the production of different castes in *R. speratus*, including primary reproductives and both brachypterous and apterous neotenics. While this mechanism seems to apply under controlled rearing conditions in the lab, its predictions do not hold up in field colonies [21]. Thus, the ability of this model to explain neotenic production in the field is suspect.

Recently, Matsuura et al. [21] proposed a genomic imprinting model to explain differentiation of the reproductive caste in termites, including *R. speratus*. According to the model, both genetic imprinting and environmental factors (e.g., the presence of inhibitory pheromones) influence the differentiation of reproductive castes. The model also predicts that the relative strength of maternal and paternal imprinting varies among species and accounts for the differentiation of both sexually produced neotenics and asexually produced neotenics (discussed below). This model, while appealing, requires confirmation, especially regarding the evidence of sex-specific epimarks. Clearly, our understanding of the proximate factors underlying neotenic differentiation in termite colonies is far from complete and needs much additional research.

Over the last decade, there have been increasing reports of a new type of extended families known as asexual queen succession (AQS). Under AQS, the founding primary queen produces neotenic queens parthenogenetically. The parthenogenetic daughters mate with the primary king to which they are unrelated, or less commonly with a neotenic brother, to produce workers, soldiers and alates through normal sexual reproduction. This unique breeding system was first reported by Matsuura et al. [25] in the subterranean termite *R. speratus* and has since been reported in two other species of *Reticulitermes* [26,27] as well as three species of higher termites [28,29]. AQS is discussed in more detail below, under Parthenogenetic Reproduction.

### 3.3. Mixed Family Colonies

Colonies in which the offspring show more complex genotypes than is possible if they are all descended from a primary reproductive pair are called mixed families. These can form either at the founding stage if multiple same sex reproductives cooperate to found colonies or later in the colony life cycle by the merging of two or more colonies. Such colonies will have five or more alleles at one or more nuclear loci and may have multiple mitochondrial DNA haplotypes present if there are two or more unrelated female reproductives present.

Mixed family colonies occur frequently in some groups of termites. Goodisman and Crozier [11] found that 5 of 19 (26%) *M. darwinienisis* colonies genotyped were mixed families. The colonies of *Zootermopsis nevadensis* (Archotermopsidae) undergo frequent colony fusion early in their life cycle. This process and the conditions leading up to it are described in detail by Thorne et al. [6,30] and Johns et al. [31]. This species founds colonies through large synchronous mating flights in which kings and queens pair up and take residence in large downed trees. The presence of many founding pairs in the same wood resource leads to high densities with closely packed nests. Interactions among neighboring incipient colonies are common and often result in the death of one of the primary reproductives and subsequent merger of the interacting colonies. Individuals of the merged colonies cooperate and function as a single colony. Colonies that engage in aggressive interactions often generate neotenic reproductives as well as reproductive soldiers. These reproductive soldiers are only aggressive against non-nestmate reproductives. Genetic studies in the lab show that reproductives from different colonies can coexist and interbreed to form mixed family colonies, and studies of mature field colonies also found the presence of mixed family colonies in this species [32]. These observations led Thorne et al. [6] to propose the “accelerated inheritance” hypothesis for the evolution of eusociality in termites. This hypothesis proposes that nonreproductive helpers (workers) that stay in the nest to assist their parents have a better chance of developing into a reproductive and inheriting the nest following aggressive encounters with conspecific colonies than they have of founding their own colony independently. These authors also hypothesize that soldiers evolved in termite colonies as an aggressive reproductive caste first and then assumed the role of sterile colony defenders later. Evidence of reproductive soldiers in other taxa of basal termites beyond the six species where they have been found would lend more credence to this idea.

Mixed family colonies have been found in the drywood termite, *Cryptotermes secundus* (Kalotermitidae), where they make up about 25% of all mature colonies studied [33]. These colonies form from the fusion of conspecific colonies when nest density is high in the same wood resource. Mixed families have been reported from subterranean termites (Rhinotermitidae) where they presumably occur through colony fusion [1,34]. Mixed family colonies of higher termites have also been reported. Multiple reproductives sometimes cooperate to found the colonies of *Macrotermes michaelseni* and maintain stable associations throughout the colony’s life cycle [35,36,37]. The colonies of *Nasutitermes corniger* often have multiple primary reproductives [13,38,39]. Studies of populations of *N. corniger* in Panama indicate that unrelated primary reproductives can arise through multiple primary reproductives associating during colony foundation, colony fusion and adoption of unrelated reproductives into established nests [13,39]. Thus, this one species forms mixed family colonies through all possible routes.

In the higher termites, mixed family colonies do not always involve primary reproductives. Haifig et al. [40] found unrelated neotenic reproductives co-existing in a colony of *Silvestritermes euamignathus*. Using mitochondrial DNA and microsatellite markers, these authors showed that the 59 neotenic reproductives found in the same nest descended from at least four different reproductive pairs representing at least two maternal lineages. Haifig et al. [40] proposed that these neotenics either descended from a group of cooperating primary reproductives or were the result of colony fusion.

The ultimate and proximate factors favoring colony fusion are not well understood. In theory, colony fusion should be selected against since it will decrease the genetic relatedness among nestmates and, therefore, decrease indirect fitness benefits that individuals might gain through helping kin reproduce. In fact, genetic studies have shown that the colonies of *R. flavipes* that fuse together are not closely related based on nuclear microsatellite markers [34]. It may be that, in some cases, the cost of defense and not fusing outweighs the cost of fusing. For example, as discussed earlier, the incipient colonies of *Z. nevadensis* can form at very high densities in down trees such that contact with other incipient colonies is inevitable as colonies grow and expand their nests [6,30]. In such situations, it may be that the cost of defense against other colonies becomes higher than the decrease in fitness due to fusion. Moreover, by merging with nearby colonies, the fused colonies are larger and likely outcompete other incipient colonies.

In terms of proximate mechanisms underlying colony fusion, there may be a number of factors involved. There is evidence of a maternally inherited factor in *R. flavipes*. DeHeer and Vargo [34] documented eight cases of colony fusion in this species, and in all cases, the colonies that fused had identical or nearly identical mitochondrial DNA haplotypes despite the fact that the neighboring colonies of this species have very different mtDNA haplotypes and despite the fact that the fused colonies were unrelated to each other at nuclear markers. The nature of this maternal factor is not known, but it could possibly be related to microbial symbionts that may be maternally inherited. For example, Matsuura [41] was able to influence the acceptance of foreign workers by manipulating the gut symbiont community in *R. speratus*. Certainly, the ultimate and proximate mechanisms underlying colony fusion in termites warrant much future study.

### 3.4. Parthenogenetic Reproduction

Thelotokous parthenogenesis has been known to occur in the eusocial Hymenoptera for many decades. It is most common in the ants where it has evolved at least four times independently and is present in at least 14 species [42]. Since first reported by Light in 1944 [43] for the genus *Zootermopsis*, the ability to produce parthenogenetic eggs has been reported from some 11 termite species in four families [44]. However, these studies primarily involved laboratory observations of either neotenic or primary queens kept isolated from males, and Light [43] concluded “there is no evidence that the ability to reproduce parthenogenetically might be of significance in the life of the species”. This all changed nearly a decade ago with the discovery of asexual queen succession (AQS) in the subterranean termite *R. speratus*, a system in which the primary queen is succeeded by parthenogenetically produced neotenic queens [25]. In the AQS system, colonies are founded in the standard way of termites where a primary king and queen pair up after a mating flight, mate and rear sexually produced offspring. The primary queen is outlived by the primary king. Before she dies, the primary queen produces a number of parthenogenetic daughters that develop into brachypterous neotenics who then mate with the king to produce workers and alates through normal sexual reproduction. Subsequent generations of neotenics are produced parthenogenetically.

AQS would seem to have many advantages over the usual production of neotenics through sexual reproduction. First, since the king is unrelated to the neotenic queens, no inbreeding takes place in the colony as long as the primary king remains alive and reproductively active. Workers and alates produced in the colony retain their genetic diversity. Second, the number of neotenic queens can be quite high, as many as 676 [45], and their combined reproductive output greatly exceeds that of a single primary queen. Thus, AQS colonies can grow faster and larger than those headed by a single monogamous pair of reproductives. From the queen’s perspective, she can continue to contribute her full complement of genetic material to future generations unlike a queen that is succeeded by sexually produced neotenics who have only half of her genes. Third, in the cases where parthenogenesis occurs by automixis with terminal fusion, i.e., in which the second polar nucleus fuses with the oocyte after the second meiotic division, as in *R. speratus* [25], the parthenogenetic daughter queens should be nearly homozygous at all loci. In such cases, genetic purging should occur through the unmasking of lethal recessive alleles in the parthenogens [45]. By producing workers and alates through sexual reproduction, these individuals retain genetic diversity which is advantageous in dealing with environmental contingencies they may face while foraging or dispersing. Neotenics, on the other hand, are sheltered and cared for by workers so that they are not disadvantaged by the lack of genetic diversity.

Since the first report of AQS in 2009, it has been found in five other species, two subterranean termites, *R. virginicus* [26] and *R. lucifugus* [27], and three higher termites, *Embiratermes neotenicus* [28], *Silvestritemes minutus* [29], and *Cavitermes tuberosus* [46]. The distribution of AQS within the termites suggests that it may have arisen independently in all of the known cases. The three *Reticulitermes* species all occur on different branches within the genus, corresponding to an Asian branch (*R. speratus*), a European branch (*R. lucifugus*) and a New World branch (*R. virginicus*) [47], and the four termitid species belong to three different subfamilies. In the end, it may turn out that AQS is more widespread than we currently know and this may lead us to revise our understanding of its evolutionary origins.

Interestingly, different modes of parthenogenesis are used by the different species with AQS, leading to different levels of heterozygosity among the parthenogens. The three species of *Reticulitermes* undergo automixis with terminal fusion, as discussed above. This results in individuals that are completely homozygous except for any crossing over that may have occurred before meiosis. Complete homozygosity should result after several generations. *Embiratermes neotenicus* undergoes automixis with central fusion, in which the two central polar nuclei fuse after the second meiotic division to restore diploidy [28]. In this case, females have identical genotypes to the queen, except at loci where crossing over occurs. This seems to be the mechanism of parthenogenesis in the social Hymenoptera (bees and ants) where thelotoky has been demonstrated [42]. Transition to homozygosity is very slow under this system. Finally, *C. tuberosis* undergoes automixis with gamete duplication [43], in which the haploid egg cell divides and the cleavage nuclei fuse to restore diploidy or the egg cell fails to divide following chromosome replication. This leads to individuals that are completely homozygous within a single generation.

Given that female reproductives in AQS species can produce both sexual and asexual offspring, how do they control which eggs are fertilized and which eggs are not? In the Hymenoptera, fertilized eggs develop into females and unfertilized eggs develop into males [48], and queens can control whether sperm is released by the spermatheca as the eggs are laid [49]. However, termites lack the ability to regulate release of sperm during oviposition. Yashiro and Matsuura [50] showed that eggs laid by the termite *R. speratus* differ in the number of micropyles, the pores through which sperm can penetrate the egg. These authors found that some eggs lacked any micropyles and these eggs developed parthenogenetically. The micropyle number changes with the age of queens, with older queens producing fewer micropyles and a higher frequency of eggs lacking micropyles. Thus, queens appear to produce asexual offspring under certain physiological conditions where they lay eggs lacking micropyles. The physiological mechanisms regulating the formation of micropyles and the molecular pathways underlying these mechanisms are yet to be determined.

Under AQS, parthenogenetically produced daughters are biased to develop into neotenics. How did these two phenomena become linked? Nozaki et al. [51] provide evidence that termites may have a preadaptation to the linkage between parthenogenetic reproduction and neotenic development. These authors set up colonies of the non AQS species *R. okinawanus* headed by two female alates. Such female–female pairs laid eggs, and although their eggs had a much lower hatch rate than those of *R. speratus*, an AQS species, the successfully produced parthenogens were homozygous and biased to develop into neotenics; one-third of parthenogens developed into neotenics compared to none of the sexually produced females. Apparently, homozygosity, a result of parthenogenesis via automixis with terminal fusion, somehow canalizes females to develop into neotenics, and this effect of parthenogenesis on developmental fate predates the evolution of AQS.

In a surprising discovery, Yashiro et al. [52] found populations of all female colonies in the drywood termite *Glyptotermes nakajimai*. This species occurs along coastal regions and nearby islands in southern Japan. The authors found that 6 of 10 surveyed populations were comprised of asexual colonies (n = 2–15 colonies sampled), whereas four populations had only sexual colonies. In asexual colonies, all workers and soldiers were females, there were no kings and queens lacked sperm in the spermatheca. There were often multiple primary queens in these colonies, up to 16, and many had from 1 to 5 neotenic queens. In contrast, sexual colonies had both male and female workers and soldiers; kings were present and queens were inseminated. Interestingly, most colonies of the sexual lineage also had multiple primary queens and often multiple primary kings, and many had multiple secondary kings and queens. The authors conducted a phylogenetic analysis and concluded that the asexual lineage arose a single time from the sexual lineage some 14 million years ago. The asexual colonies had fewer soldiers, but the soldiers were more uniform in their headwidth, leading the authors to conclude that the soldiers in the asexual lineage were more efficient in their defensive role. Primary reproductives of the sexual lineage are capable of producing some viable asexual eggs, but these hatched at a much lower frequency than sexual eggs from this lineage or the asexual eggs of the asexual lineage.

Yashiro et al. [52] hypothesized that there were a number of traits in the ancestors of the asexual lineage of *G. nakajimai* that pre-adapted it to the evolution of all female societies. The first is the occurrence of tachyparthenogenesis, the development of some unfertilized eggs laid by normally sexually producing females. The second trait is pleometrosis, cooperative colony founding by queens, which appears to be common in the sexual lineage. By grouping together, termite queens can survive colony foundation through mutual grooming to avoid disease infection. Finally, the authors suggest that being a single piece nester, as is true of all drywood termites and many other termites, individuals do not have to leave the nest to forage for food, reducing the risk of exposure to many environmental factors, such as pathogens, parasites and other natural enemies, in which genetic diversity may be advantageous. Without the biotic pressure to reshuffle the genome through sexual reproduction to better deal with natural enemies, selection may have favored all female colonies, resulting in higher fitness of the female reproductives. The discovery of an asexual lineage in one termite species certainly opens the door to finding additional cases in other termites.

### 3.5. Ecological and Evolutionary Factors Shaping Termite Breeding Systems

The large variation in colony breeding structure within and among species [2] raises the question of the selective forces favoring one breeding system over another, for example, when are extended families favored over simple families, or when are mixed families favored over extended families? Vargo et al. [24] studied several populations of *R. flavipes*, along its range in the eastern U.S., and *R. grassei*, in its range in the Iberian Peninsula and southwestern France. In *R. flavipes*, these authors found a negative relationship between latitude and both the proportion of colonies that are simple families and the level of inbreeding in colonies. That is to say, populations in the southern portions of its range had mainly simple family colonies and those in the northern portion had mainly extended families (see Figure 1). Correspondingly, populations in the southern part of the range were less inbred than those in more northern locations. *R. grassei* showed a similar relationship, but only the increase in inbreeding with latitude was significant. Colonies that produce neotenics can replace their primary reproductives when they die. Thus, differences among colonies in their breeding system are largely due to age. Clearly, colonies in the southern portion of the ranges of these species either do not live long enough to produce neotenics, or they do not live long after neotenics develop within the colony, possibly because the inbred colonies are disadvantaged compared to the outbred colonies.

To investigate the possible role of local environmental conditions in shaping colony breeding structure, Vargo et al. [24] examined several bioclimatic factors in relation to colony breeding structure and the level of inbreeding in *R. flavipes* and *R. grassei*. In both species, the degree of inbreeding was higher in cooler, moister habitats. In *R. flavipes*, inbreeding was most strongly affected by climatic conditions (mean annual temperature and seasonality), whereas in *R. grassei*, biological variables, such as the availability of wood resources and soil composition, were most closely associated with inbreeding. These results suggest that local ecological conditions may be important in influencing colony breeding structure, but how they favor one breeding system over another is not understood.

In the higher termite *Macrotermes michaelseni*, Brandl et al. [36] found geographic variation in the frequency of polygyny (multiple queen colonies); colonies at the edge of its range were more likely to be polygynous. The authors suggest that it is more difficult for monogamous pairs of reproductives to found colonies near the border of their range due to ecological constraints, and groups of female primary reproductives have a better chance of founding colonies successfully under such conditions. Clearly, a more detailed understanding of the ecological factors influencing colony breeding structure is an important area for future studies.

## 4. Conclusions

There is tremendous diversity in the breeding system of termites both within and among species. In addition, the breeding system within a colony may change as the colony ages; most often, this involves a colony transitioning from a simple family to an extended family following the death of one or both of the primary reproductives and the production of replacement neotenics. It can also occur when two or more colonies fuse to form a mixed family colony. Thanks to molecular genetic techniques, there is an increasing number of studies on the breeding systems of termites yielding more detailed variation about how colonies within a population differ and how populations within a species can vary. One of the most exciting new developments is the discovery that several termite species, including both lower and higher termites, use parthenogenetic reproduction as an integral part of their lifecycle. In the coming years, there are sure to be many more species found that use asexual reproduction. Continuing to document the range of breeding systems in termites using a combination of field collections and genetic studies will remain an important area of research.

Although our understanding of the variation in breeding systems has been rapidly increasing, our knowledge of the evolutionary and ecological factors responsible for this variation lags far behind. Studies are needed to investigate the relationship between breeding system and ecological conditions, such as temperature, availability of food resources, population density, pathogen pressure, etc., as well as life history attributes such as nesting habits, food resources utilized (wood, soil, fungus) and phylogenetic relationships. In addition, laboratory and field studies investigating the possible deleterious effects of inbreeding are also needed to better understand the costs and benefits of inbreeding in extended family colonies or the lack of inbreeding under AQS. Variation in breeding systems is common throughout social insects. Termites provide an excellent group for providing insights into the forces shaping breeding systems in social insects.

## Figures and Tables

**Figure 1 insects-10-00052-f001:**
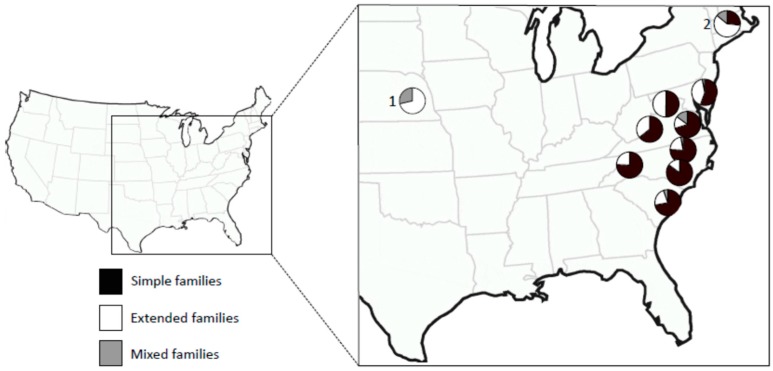
Variation in the breeding system in populations of *Reticulitermes flavipes* in the U.S. as determined by genetic markers. ^1^ Data from DeHeer and Kamble [22]; ^2^ data from Bulmer et al. [23]; all other data from Vargo et al. [24]. Sample sizes were between 7 and 314 colonies per population.

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
