# Peer review of "Diversity of Termite Breeding Systems"

_insects, 2019, doi:10.3390/insects10020052_

Round 1
Reviewer 1 Report
This review has the clarity and competence one would expect from the author. Reviews should be evaluated with a light touch, as after all they are perspectives and not additions of new data and specific interpretations. So I list the following minor issues:
p2, line 14. delete "of the termite breeding systems".
p2 lines 15-22. A bit repetitious. Could these definitions be integrated into the first paragraph?
p2 line 27. The main reproductive caste in termites is the primary reproductive? Male and female primaries are not generally seen as separate castes, or are they?
p2 line 41 and elsewhere. What is a group? Does this imply they are housed together, or merely constitute a subclass of individuals with a number of members?
p3 lines 17-23. Reads slightly awkwardly, specially where "replace" is used twice in the same sentence.
p3 line 28 and elsewhere. R presumably means Reticulitermes, but this is not previously specified. Once specified it should remain abbreviated throughout, except where the characteristics of the genus in general are discussed.
p5 line 10. What does "same sex" mean? Surely only females can behave in this manner?
p3 line 29. "posits"? OK, it's in he dictionary, but isn't "proposes" a more modern term?
p, line 49 and following. "Automixis with terminal fusion". Typo here, but more importantly should this term be defined here (it is partly so on the next page), as in p7, lines 16-22.
p8 line 27. " … additional cases in other termites".
Author Response
Point 1: p2, line 14. delete "of the termite breeding systems".
Reply: I have followed the reviewers advice.
Point 2: p2 lines 15-22. A bit repetitious. Could these definitions be integrated into the first paragraph?
Reply: I have rewritten par. 1 to take into account the reviewer's comments.
Point 3: p2 line 27. The main reproductive caste in termites is the primary reproductive? Male and female primaries are not generally seen as separate castes, or are they?
Reply: I have clarified this to indicate male and female primary reproductives are different castes.
Point 4: p2 line 41 and elsewhere. What is a group? Does this imply they are housed together, or merely constitute a subclass of individuals with a number of members?
Reply: I have used the term aggregation to clarify this point.
Point 5: p3 lines 17-23. Reads slightly awkwardly, specially where "replace" is used twice in the same sentence.
Reply: I have replace one incident of "replace" with "supersede."
Point 6: p3 line 28 and elsewhere. R presumably means Reticulitermes, but this is not previously specified. Once specified it should remain abbreviated throughout, except where the characteristics of the genus in general are discussed.
Reply: I have taken care to use only R when possible.
Point 7: p5 line 10. What does "same sex" mean? Surely only females can behave in this manner?
Reply: I'm not sure which incident use of "same sex" the reviewer is referring to, but in both cases where I use the term both males and females can behave this way.
Point 8: p3 line 29. "posits"? OK, it's in he dictionary, but isn't "proposes" a more modern term?
Reply: I changed the text.
Point 9: p, line 49 and following. "Automixis with terminal fusion". Typo here, but more importantly should this term be defined here (it is partly so on the next page), as in p7, lines 16-22.
Reply. I followed the reviewer's advice and changed the wording.
Point 10: p8 line 27. " … additional cases in other termites".
Reply: Correction made.
Reviewer 2 Report
This is a well written, comprehensive but descriptive Review manuscript, where the author goes through the diversity of breeding systems in termites. I feel it is much in the style of a text book, and I miss the personal input of the author and his experience in the field (apart from the self-citations). Basically, I feel the need of more clearly hypothesis stating. Research suggestions and relevance of the different research lines possible could be critically evaluated. However, and with some exceptions, I feel that the current state of understanding on the topic is well presented.
I am attaching the annotated pdf with specific comments, but I would like to call the attention to three points that I would like to see better discussed:
- The statement on page 5: “Mixed families have been reported from subterranean termites (Rhinotermitidae) where they presumably occur through colony fusion [1, 31].” Several more are the studies where, at least for Reticulitermes, mixed families have been reported and related with colony interactions and/or with the invasive character of the species. A great majority of these studies consider European Reticulitermes species. Potencial drivers of colony fusion could be hypothesized here in a sustained matter. Those are also not discussed on page 6, where cost/benefits and the mechanisms underlying colony fusion are referred.
- The particular case of fungus-growing termites is largely ignored in the all review (with the exception of a small sentence on M. michaelseni). There is indeed not much information, but they should also be discussed. The singularity of having an external obligatory symbiont – Termitomyces – that is reared in a monoculture and is either vertical or horizontally transmitted might be (or not) an extra conditioning of the family structure.
- It was briefly referred on page 6 for R. speratus the potential relevance of the symbiont community on colony fusion, but the fact that termites are in its great majority, in a way or another, entangled in obligatory symbiosis is never highlighted nor its’ potential to shape behavior and eventually breeding system referred to. Particular, the more iconic examples of lower termites dependency of protists or the fugus-growing termites dependency of Termitomyces could/should be more explored.

Author Response
Point 1: The statement on page 5: “Mixed families have been reported from subterranean termites (Rhinotermitidae) where they presumably occur through colony fusion [1, 31].” Several more are the studies where, at least for Reticulitermes, mixed families have been reported and related with colony interactions and/or with the invasive character of the species. A great majority of these studies consider European Reticulitermes species. Potencial drivers of colony fusion could be hypothesized here in a sustained matter. Those are also not discussed on page 6, where cost/benefits and the mechanisms underlying colony fusion are referred.
Reply: I cite a paper dealing with invasive populations of R. flavipes. I don't really understand what the reviewers' point is. I feel I have addressed the major hypotheses that have been proposed.
Point 2: The particular case of fungus-growing termites is largely ignored in the all review (with the exception of a small sentence on M. michaelseni). There is indeed not much information, but they should also be discussed. The singularity of having an external obligatory symbiont – Termitomyces – that is reared in a monoculture and is either vertical or horizontally transmitted might be (or not) an extra conditioning of the family structure.
Reply: I discuss what we know about variation in the breeding structure in this group. I don't think it's appropriate for me to speculate on why there is so little variation about this group in this review.
Point 3:It was briefly referred on page 6 for R. speratus the potential relevance of the symbiont community on colony fusion, but the fact that termites are in its great majority, in a way or another, entangled in obligatory symbiosis is never highlighted nor its’ potential to shape behavior and eventually breeding system referred to. Particular, the more iconic examples of lower termites dependency of protists or the fugus-growing termites dependency of Termitomyces could/should be more explored.
Reply: I think the reviewer is being highly speculator and I don't think it is appropriate for me to go down this path in this review.
Reviewer 3 Report
This informative manuscript reviews current understanding of breeding systems within and among termite species. New findings regarding asexual queen succession are discussed, and the author identifies areas of research that should be explored further. Research into reproductive strategies among termites has revealed how much remains to be learned about this intriguing subject. Both environmental factors and colony age can influence termite breeding systems, which are highly variable and flexible. This manuscript is a well-written and comprehensive overview of the state of our knowledge about the subject and it will be very helpful to researchers deciding which question to tackle next. There were only a few errors in the text, including (1) p.2, second to last line, should be either "group far exceeds.." or "group can far exceed...", (2) p.3, line 6 under 3.1, reproductives is misspelled, and (3) p.4, line 7 on third paragraph, it should be R. santonensis, not R. flavipes.
Author Response
Point 1: (1) p.2, second to last line, should be either "group far exceeds.." or "group can far exceed...",
Reply: Change to "far exceed"
Point 2 (2) p.3, line 6 under 3.1, reproductives is misspelled
Reply: corrected
Point 3: (3) p.4, line 7 on third paragraph, it should be R. santonensis, not R. flavipes.
Reply: R. santonensis has been synonomyzed with R. flavipes.